# Implementing Modern Technology for Vital Sign Monitoring to Enhance Athletic Training and Sports Performance

**Răzvan-Sandu Enoiu, Iulia Găinariu *** and **Veronica Mîndrescu**

Faculty of Physical Education and Mountain Sports, Transylvania University Brasov, 010374 Brasov, Romania
* Correspondence: iulia.gainariu@student.unitbv.ro

**Abstract:** (1) Background: Applying medical procedures to keep track of athletes' metabolic health is a well-known behavior for increasing sports performance. This study's primary goal was to examine whether implementing a health-screening routine using invasive and non-invasive methods in a mountain running training program can increase sports performance by obtaining a better rank in specific races. (2) Methods: Seven male mountain runners with good results at the international level participated in the research, which featured an initial and a final test. The initial test took place in March at the beginning of the competitive season, and the final test took place in September at the end of the annual training plan in 2021. The entire experiment used invasive tests, such as taking blood samples to perform blood biochemistry analysis, and non-invasive tests, such as mountain running races, determinations of $VO_2max$, EKGs, urine biochemistry and NeuroTracker tests. (3) Results: Comparing the initial test with the final one from a statistical point of view, a significant improvement was found in the final test regarding the obtained results in most tests and, most importantly, the occupied position in the final general rank ($p = 0.005$). The main variations after effort were decreased $Ca^{++}$ and increased $Cl^{-}$, a low TGL level if the diet was inadequate, increased LDH and CPK strongly correlated with the skeletal muscle response, and only physiological changes were found in the EKG and urine summary. (4) Conclusions: The invasive and non-invasive tests applied in this study provided crucial information on the athletes' health statuses, enabling the coach to adjust the training program in light of the findings in order to improve sports performance and avoid overexertion.

**Keywords:** performance; technology; prevention; $VO_2max$; NeuroTracker

## 1. Introduction

Mountain running is one of the most recent events in athletics, in which the metabolic and neuromotor demands of athletes are at a maximum [1]. Due to the increasing popularity of this sport (López-García et al., 2022) [2], as well as its novelties, this research proposed to investigate a series of biochemical and neuromotor parameters in mountain runners using invasive and non-invasive techniques for monitoring athletes' performances and health statuses. Interpreting the biochemical indicators used in this study allowed for the direct examination [3] of each athlete's health state and level of exhaustion, making it feasible to modify physical training in accordance with these factors [4].

Since the national senior mountain running team has had a remarkable development in recent years [5], this study was indispensable to the development of this sport on the national plan and for obtaining scientific information regarding their biochemical, cardiological and psychological status in order to provide valuable information to their coach that can be applied in their training program.

Starting from the theoretical premise of the fact that these parameters are useful for screening the health statuses of athletes, a series of invasive tests [6] were used in the research, which were represented by the hematological analysis of blood, the analysis of blood gases and the determination of the lactate concentration in the blood capillaries [7]; a series of non-invasive tests represented by the identification of the $VO_2max$ value through

the effort test carried out on the treadmill [8], a mountain running race, performing an electrocardiogram, carrying out specific training through the NeuroTracker device and a urine summary [9] were also performed. These tests were selected with the purpose of detecting a minimum of metabolic, cardiological, renal and psychological changes that occurred after specific competition tests, but which provided a complete profile that reflected the real health status of the subjects and that can predict their sports performances [7].

Thus, the specific sports test can quantify the individual sports performance [10] of the subjects under research, while the VO$_2$max value reveals the instantaneous effort capacity of each athlete during maximal effort and is a reliable indicator of cardiorespiratory health [8]. The physiological adaptations of the heart to prolonged physical training produce electrocardiographic changes that are considered abnormal in untrained people [11] but are perfectly physiological for high-performance athletes [12]. For the early detection of silent cardiovascular diseases, the European Society of Cardiology and the International Olympic Committee demonstrated the importance of the 12-lead electrocardiogram (EKG) as part of routine screening during physical training and before participation in competitions [13]. Since one of the most formidable tasks for a mountain runner's brain during training is to perceive and integrate complex movement patterns in real time, which requires attention and unlimited information resources, the inclusion of NeuroTracker dynamic perceptual–cognitive training in the process of training can improve the perfectly trainable perceptual–cognitive abilities of the human brain, thus becoming an indispensable component in increasing sports performance [14]. The obtained information from the microscopic examination of the urinary sediment is an appropriate method of screening or diagnostics because it highlights the changes that can occur in certain pathological states of the urinary system, metabolism and liver [15]. Since it is an easy, simple and non-invasive test with high sensitivity and specificity, the urine summary is routinely used to monitor the health status of athletes [9].

All physiological functions in the body typically depend on maintaining the acid–base balance [16]. Interstitial and intracellular pH are influenced by the arterial blood pH, which ranges from 7.35 to 7.45 under normal physiological circumstances. The pH-dependent enzymes and membrane transport proteins do not function correctly if pH varies from its typical range [17]. The metabolic pathways may be significantly impacted in these circumstances, which will change physical performance [18]. Two physiological processes—namely, the physiological buffer systems and physiological regulation via the pulmonary then renal system, which has a greater trigger mechanism—are in charge of steadily but slowly regulating the acid–base balance [19]. In essence, buffer systems prevent the pH from rising or falling by forming a mixture of substances that resist pH change [20]. However, in particular situations, such as during long and strenuous physical workouts, the buffer systems may not be sufficient [21]. Hematological blood testing and gas analysis are two methods that can help athletes avoid metabolic acidosis and other electrolyte imbalances that can adversely affect their performance on the field of play and their overall health over the long term [22].

Blood lactate concentration is one of the most widely used parameters measured during sports training because these determinations can prescribe appropriate effort intensities for an athlete's training [23]. Although the utility of this method is increased, the interpretation of the data obtained can be a challenge because dramatic increases in lactic acid may also characterize a normal response to intense exercise if the rate of lactate elimination is lower than the rate of entry into the blood [24]. The increased values do not necessarily signify hypoxemia or ischemia. However, the ability to maintain an acid–base balance is one of the main factors limiting physical capacity [25].

Thus, one of the main objectives of this research was to determine the applicability of these selected invasive and non-invasive tests [19] in practice and to establish their effectiveness during an annual training plan. Since these tests bring important information about the general state of an athlete's health [26] and can characterize the metabolic, cardiopulmonary and psychological responses of athletes to effort [27], their use within a

well-designed protocol can direct sports training and lead to an increased effort capacity, avoid overfatigue and accidents, and maximize sports performance [28]. Furthermore, an important aspect of obtaining and maintaining the athletes' performance is to adopt well-balanced and adequate nutrition [29].

## 2. Materials and Methods

### 2.1. Participants

All the participants in this experimental study participated by invitation. The experimental survey was conducted between March and September 2021. The seven elite mountain running male athletes who participated in this study undertook the initial test at the start of the annual training plan in March and the final test in September in order to emphasize the dynamics of the recorded data. The inclusion criteria of the subjects in the research were that all participants should be free of any disease, have very good athletic achievements at the national level and be members of the national team trained by the same athletics coach.

The research was realized at the National Research Institute for Sports in Bucharest and the mountain running races were performed through the National Mountain Running Championships, specifically the first stage and final stage, in Câmpulung Moldovenesc.

This study was approved by the Ethics Review Committee of the Transilvania University from Brasov and all the subjects were informed orally and in writing about the purpose and methods of this study. We obtained their written consent to participate in the study after they fully understood the content of the study.

The participants in the study were older than 20 years and underwent the anthropometric measurements listed in Table 1.

**Table 1.** The athletes' anthropometric data.

| Athlete | Height (cm) | Weight (kg) | Age (2021) (Years) | BMI (kg/m$^2$) |
|---------|-------------|-------------|--------------------|----------------|
| A1 | 176 | 67 | 26 | 21.63 |
| A2 | 173 | 65 | 32 | 21.72 |
| A3 | 175 | 67 | 34 | 21.88 |
| A4 | 178 | 58 | 38 | 18.31 |
| A5 | 176 | 70 | 28 | 22.6 |
| A6 | 175 | 50 | 22 | 16.33 |
| A7 | 170 | 62 | 20 | 21.45 |

Legend: cm—centimeters, kg—kilograms, m$^2$—meters squared.

### 2.2. Procedures

The participants undertook the initial round of testing in March 2021, starting with the first stage of the National Mountain Running Championships, and the final round of testing was done in September 2021, with the final stage of the National Mountain Running Championships. According to the Table 2, each test was conducted in accordance with a well-established protocol to prevent interference with the results. This included a specific warm-up that respected the accommodation period on the treadmill, a complete cool-down and a period of fasting before taking blood samples.

**Table 2.** Procedures applied [30].

| | Protocol Respected | Duration (min) |
|---|--------------------|----------------|
| Mountain Running Race | Warming up | 30 |
| | Cool-down | 20 |
| EKG | Performed in rest conditions | 10 |

**Table 2.** *Cont.*

|  | Protocol Respected | Duration (min) |
|---|---|---|
| Effort test | Individual gymnastics | 15 |
|  | Easy running on the stadium | 15 |
|  | Easy running on the treadmill | 8 |
|  | Bruce protocol to measure $VO_2max$ | 18–20 |
|  | Cool down (running and stretching) | 30 (15-15) |
| Lactate analysis | Blood samples were performed at the end of the running on the treadmill and 15 min later | 16 |
| Biochemical blood test | No breakfast before | 5 |
|  | Samples were performed before exercise |  |
| Astrup method | No breakfast before | 5 |
|  | Samples were performed before exercise |  |
| Urine summary | No breakfast before | 5 |
|  | Specific protocol to avoid contamination of the sample |  |
| NeuroTracker | Specific NeuroTracker protocol | According to the protocol |

Legend: min—minutes.

### 2.3. Materials and Testing

2.3.1. Mountain Running Race, $VO_2max$, Lactate and NeuroTracker Results

a. Mountain Running Race

The sports test consisted of covering a route specific to the mountain running race, which was selected to measure and objectify the individual physical capacity and performance level of the research subjects. Participation in specialized competitions is the only real test that can compare the level of sports performance achieved by each individual athlete through the obtained results in the general rank. The initial and final mountain running tests consisted of covering identical distances within the national mountain running championships, namely, the first stage of the National Mountain Running Championships on 11 April 2021 and the final stage on 19 September 2021. The running routes both measured 10.00 km and had 95% forest road but had different altitude levels. In the first race, the elevation was +453 m, and in the second race, the elevation was +947 m, according to Figure 1.

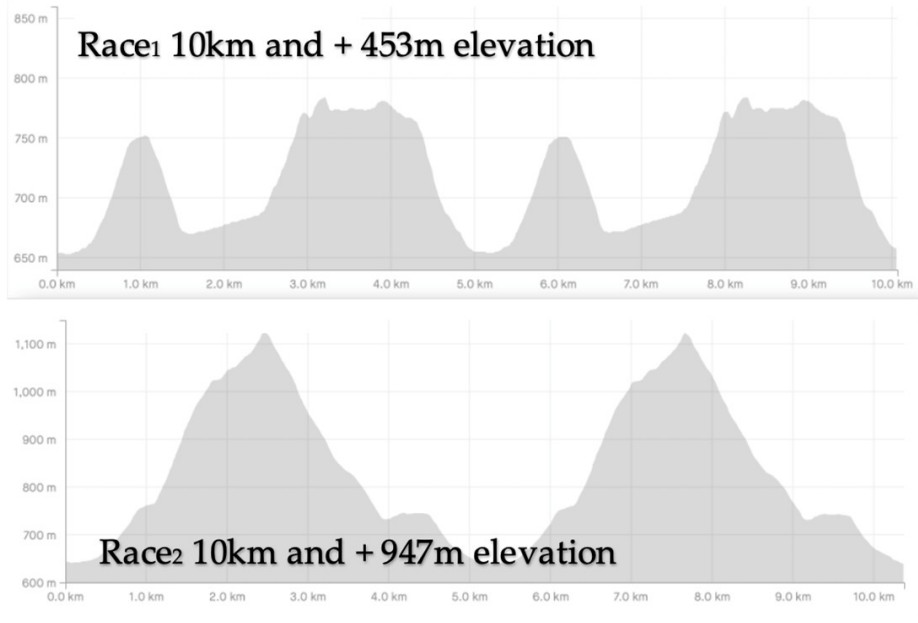

**Figure 1.** Mountain running race routes.

b. Effort test

The VO$_2$ max test was was necessary in order to understand each athlete's reaction to maximal effort because each human can have a different reaction to the same stimulus, which can be used to determine the anaerobic threshold [8]. The effort test is a noninvasive procedure that was conducted on a treadmill in a closed-circuit gym with participants connected to the fourth-generation Cosmed K5 device through a portable system. The athletes wore a backpack that included this gadget, which tracked their gas exchange, air flow and VO$_2$max and provided gas analysis (O$_2$, CO$_2$). The software of the Cosmed gadget automatically calculated the VO$_2$max values and categorized and analyzed the outcomes in an Excel table structure [31]. The following is how the findings should be interpreted in light of the device guide: values below 36.7 are regarded as extremely weak, those between 36.7 and 41.0 are regarded as moderately weak, those between 41.0 and 44.4 are regarded as acceptable, those between 44.44 and 47.5 are regarded as good, those between 47.5 and 54.3 are regarded as excellent, and anything above 54.3 is regarded as superior [8]. Trying to keep the effort using the maximum step as much as possible, all athletes underwent the Bruce protocol (Table 3), which consists in staying on each effort level for three minutes [23].

**Table 3.** Bruce protocol.

| Level | 1 | 2 | 3 | 4 | 5 | 6 | 7 |
|---|---|---|---|---|---|---|---|
| Speed (km/h) | 2.7 | 4 | 5.4 | 6.7 | 8 | 8.8 | 9.6 |
| Elevation (%) | 10 | 12 | 14 | 16 | 18 | 20 | 22 |

Legend: km—kilometers, h—hour.

c. Lactate analysis

This test was performed with the Lactate Pro2 lactate analyzer, which analyzes lactate quickly with a minimum of only 0.3 microliters of capillary blood. This method gauges the intensity of the physical exercises quickly, cheaply and accurately [32]. The first and last blood samples were obtained from the fourth finger of both hands at the end of the effort test and exactly 15 min after the treadmill activity was completed, respectively; the break in between included walking, easy jogging and stretching. The finger was initially cleaned with an antiseptic solution and dried with a cotton disc to prevent contaminating the samples. The purpose of this test was to correlate the lactate threshold reached by each athlete with the one registered in the VO$_2$max test and to detect the recovery rate.

d. NeuroTracker

Starting from the premise that the psychological component of sports training is the most underdeveloped, we supposed that training the cognitive component may improve focus and mental capacity, which, in turn, will improve sports performance. According to the NeuroTrackers users and specialists, any user who follows this procedure should notice an improvement in their concentration and attention capacity, which will boost their capabilities regarding speed, working memory and raw attention [14]. Two stages of the program, namely, 18 visits and around 42 sessions with a 10 min duration for each athlete, were carried out in this study. The exam took over 7 h to complete, which was broken up into 18 visits of roughly 23 min each visit [33].The participant was centered on the screen and kept at a distance of 1.5 m during the testing method. Each user who followed the entire procedure should considerably improve their ability to concentrate, pay attention, increase their information processing speed, and develop their working memory and raw attention [14].

2.3.2. EKG Test

An extremely crucial amount of information on the electrical activity of the myocardium can be obtained with the use of the EKG, which is a rapid, easy and non-invasive diagnostic tool. In summary, performance athletes with superior aerobic capacity have physiological adaptations to exercise that are more significant for their cardiac evaluation,

whereas sedentary people may suffer changes that might suggest heart abnormalities [34]. It is also crucial that the EKG is interpreted based on symptoms, family history and the setting of the individual, in this case, high-performance athletes. Therefore, it is preferable for a specialist with expertise in sports cardiology to evaluate these findings [35]. The International Olympic Committee and the European Society of Cardiology both strongly advise using an EKG as part of routine screening both before and during physical activity [13]. This is a precise indicator of the presence of undiagnosed cardiovascular disease. Therefore, depending on the athlete's medical history and using a comprehensive physical examination, any qualified physician can recognize the danger signs during an EKG [36]. The athletes had an EKG before and after the two mountain running competitions.

### 2.3.3. MicroAstrup, Biochemical Blood Test and Urine Test

After each mountain running competition, the initial and final testing involved the biochemical analysis of the capillary blood drawn from the pulp of the fourth finger and hematological analysis using the Medonic device and the Radiometer device to observe the biochemical changes that resulted from the prolonged physical effort. Additionally, urine samples under the same circumstances were subjected to biochemical testing to look for potential renal system disorders.

a. MicroAstrup

The invasive Astrup approach is based on the inverse connection between blood pH and the partial pressure of carbon dioxide ($pCO_2$) [37]. This test was performed with the ABL9 Radiometer automatic blood gas analyzer to extract the parameters of interest from the blood samples in just 35 s and send them automatically to a computer in an Excel format [38]. The studied parameters were the following: oxygen partial pressure ($pO_2$), carbon dioxide partial pressure ($pCO_2$), pH, bicarbonate anion ($HCO_3^-$), calcium ionic concentration ($Ca^{++}$), magnesium ionic concentration ($Mg^{++}$), chloride ionic concentration (Cl), sodium ionic concentration ($Na^+$), potassium ionic concentration ($K^+$), lactate concentration (AL) and oxygen saturation ($SO_2$). Special care had to be taken during the sample collection and handling in order to prevent any potential errors. Poor blood sampling, which might have resulted in hemolysis or venostasis, was avoided. Additionally, the sample should not be taken in the wrong sort of tube or in the wrong way; otherwise, it can be inappropriately combined with substances such as anticoagulants or preservatives in the collecting tube [39]. In order to carry out the MicroAstrup test, the blood was collected after the effort in the running competitions organized during the initial and final testing. Following the two samples of results obtained after the initial and final testing, only the values of the modified parameters are given in the results section.

b. Biochemical blood test

In this sample, a complete blood count was performed to reveal important aspects regarding the erythrocyte, leukocyte and platelet populations, including serum iron and ferritin; complete lipid profile: total cholesterol, low-density lipoprotein (LDL), high-density lipoprotein (HDL) and triglycerides (TGL); inflammation markers: C-reactive protein and creatine phosphokinase (CPK) [40]; and complete ionogram: sodium, potassium, chlorine, magnesium, serum calcium bicarbonate and phosphate.

The purpose of this test was to interpret all the medical markers below in order to detect any biochemical abnormalities, such as organs under stress, inflammation, blood disorders, anemia and infections [41]. Serial tests were used to track each athlete's health status, evaluate the success of the training sessions and determine the biochemical adaptation of the body to the effort. These tests are crucial for keeping track of the athletes' health, which allows the coach to modify the diet, nutritional consumption, rest times, training sessions and electrolyte intake according to the results with the aim of preventing overtraining and injuries [42]. Hematological analysis was performed with one of the quickest hematological analyzers in the world, namely, the Medonic device. The machine requires twenty microliters of capillary blood to perform the analysis. For reliable findings, blood samples were taken in an EDTA K3 or K2 tube before being gently mixed. The results

are shown on a touch screen in less than a minute and sent to an Excel document. The micropipette sample can be evaluated immediately after collection and, for best results, it should be performed in no more than 10 min [43]. The hematological analysis of the blood was performed after each mountain running competition within the initial testing and the final testing so that, in total, two hematological bulletins were obtained. The most relevant changes found are recorded in the Results section. The obtained results were automatically sent by the Medonic device software directly to the computer in the Excel format.

c. Urine summary

Using a noninvasive technique, urine can be collected and transported to the laboratory to be analyzed. It is crucial to produce the urine summary with the least amount of mistakes possible. In order for this to happen, only the day's first urine is suitable to collect (morning urine). The initial stream of urine should not be collected since it typically contains germs that have colonized the urethra. As a result, the middle stream of urine is collected because it contains more pee and is less likely to be polluted by the commensal flora of the urethra. The urine test helps to identify potential urinary system infections and track kidney function and glomerular filtration efficiency, which is crucial for athletes who engage in intense long-time exercises [9]. The used parameters were pH, urinary glucose, ketone bodies, nitrites, urinary proteins and biliary pigments.

These factors made it possible to directly assess each athlete's kidney health and the degree of exhaustion, giving the coach the chance to adjust the training program in correlation with these parameters in order to maximize the training efficiency and maintain the health of the renal system [19]. The urinalysis was collected after the specific effort samples during the initial and final testing.

### 2.3.4. Statistical Analysis

For the statistical analysis, a paired-samples $t$-test was performed in order to determine whether an improvement between the initial and final tests existed. The $t$-test was performed for the most suitable dependent variables: Rank1- Rank2, $VO_2maxi$-$VO_2maxTf$, NTI-NTF, $pO_21$-$pO_22$, $SO_21$-$SO_22$ and LDH1-LDH2, using the SPSS software statistics program (IBM, Romania). By obtaining the $p$-value, the null hypothesis could be rejected, and the research hypothesis confirmed whether the $p$-value was lower than the standard $p$-value of 0.05 [44].

## 3. Results

### 3.1. Mountain Running Race, $VO_2max$, Lactate and NeuroTracker Results

The mountain running races were performed during the 2021 competition year on 11th April and 19th September and the athletes obtained the results shown in the following Table 4:

**Table 4.** Comparative results between initial and final race.

| Athlete | Time1 | Time2 | Rank 1 | Rank 2 | Min1 | $VO_2$/kg1 | HR max1 | Min2 | $VO_2$/kg2 | HR max2 | LATI1 | LARI1 | LATF2 | LARF2 | NeuroI | NeuroF |
|---|---|---|---|---|---|---|---|---|---|---|---|---|---|---|---|---|
| A1 | 2481 | 3685 | 1 | 4 | 19.03 | 70.4 | 195 | 19.35 | 71.2 | 199 | 12.2 | 6.1 | 11.2 | 5.6 | 1.88 | 2.12 |
| A2 | 2500 | 3570 | 2 | 1 | 18.25 | 70.1 | 188 | 19.2 | 70.6 | 188 | 15.8 | 7.5 | 12.5 | 7.1 | 1.17 | 1.86 |
| A3 | 2513 | 3749 | 3 | 6 | 18.47 | 65.1 | 197 | 19.18 | 68.1 | 197 | 13.5 | 6.9 | 12.8 | 6.5 | 1.05 | 1.76 |
| A4 | 2559 | 3608 | 6 | 2 | 19.11 | 63.2 | 201 | 19.8 | 64.2 | 201 | 12.8 | 6.5 | 11.9 | 6.1 | 1.41 | 1.88 |
| A5 | 2564 | 3669 | 7 | 3 | 18.1 | 52.8 | 205 | 19.15 | 54.6 | 210 | 12.1 | 7.1 | 11.7 | 6.8 | 1.86 | 2.09 |
| A6 | 2692 | 4028 | 12 | 8 | 18.51 | 43.5 | 198 | 19.1 | 48.7 | 198 | 14.7 | 10.8 | 12.8 | 9.9 | 2 | 2.52 |
| A7 | 2908 | 4409 | 16 | 9 | 19.17 | 64.4 | 200 | 19.23 | 65.1 | 200 | 12.1 | 5.9 | 12 | 5.5 | 1.67 | 2.31 |

Legend: Time1—time spent during the initial race (seconds), Time2—time spent during the final race (seconds), Rank1—rank in the first race (general classification), Rank2—rank in the second race (general classification), Min1—Bruce protocol duration (minutes) vs. Min2—Bruce protocol duration (minutes), $VO_2$/kg (mmol/L) for the initial and final races, HRmax for the initial and final races, LATI1—initial test lactate (mmol/l) vs. LARI1—initial rest 1 lactate (mmol/L), LATF2—final test lactate (mmol/L) vs. LARF2—rest test 2 lactate (mmol/L), NeuroI and NeuroF—NeuroTracker test scores for the initial and final races.

Since there was a difference in altitude level between the two mountain running routes used as the sport test during the initial and the final tests, the times achieved also differed, where given the altitude difference in the final test was higher, the duration was expected to be longer. For this reason, in order to compare the obtained results, the occupied positions in the general ranks of both competitions were used.

In both competitions, three out of seven athletes obtained the first three positions in the national general rank, a fact that denotes the good methodology applied by the coach in the training process and the superior physical capabilities of the athletes. While the best-ranking position, namely, first place, was maintained, the lowest-ranking position obtained by A7, namely, 16th place in the initial test, went up seven ranking positions in the final test.

The calculations of the VO$_2$max values were performed automatically by the Cosmed device's software, which classified and interpreted the results directly in an Excel table system. The interpretation of the results in relation to the device guide classifies the results as follows: values below 36.7 are considered very weak, between 36.7 and 41.0 is considered weak, between 41.0 and 44.4 is considered acceptable, between 44.44 and 47.5 is considered good, between 47.5 and 54.3 is considered excellent and everything above 54.3 is considered superior [45]. The VO$_2$max test revealed a high level of aerobic capacity for all the tested subjects, and each of them reached the last step of the Bruce protocol. All subjects exceeded their maximum initial time performed on the treadmill within the protocol, staying on the last step of the Bruce protocol and obtaining VO$_2$max values that were superior to the initial testing. The myocardial speed contraction was higher in the case of two out of seven subjects, but all subjects had a heart rate toward the maximum limit.

Regarding the lactate concentrations, better results were obtained in the final test, with the lactate production being slightly lower compared with the initial test, although the effort performed was at least as intense as the initial test and the obtained results were higher. The lactate recovery rate was higher in the final testing for three subjects, while it remained identical for one subject and it was slightly lower for three subjects.

*3.2. EKG Analysis Results*

An EKG is a non-invasive, simple-to-use test that can detect a serious cardiac issue that was missed during the patient's history and physical examination at a reasonable cost. Through these tests, precise information on the heart's electrical activity, including p-wave changes, QRS complex, ST segment and T wave, as well as the number of beats per minute (BPM), was obtained. More than 50% of athletes in the study exhibited EKG pathway alterations, which indicated typical problems [35].

However, the interpretation of the differences occurring within the group of athletes regarding the electrical changes of the myocardium obtained through the EKG analysis (Table 5) demonstrated the individual's specific cardio-physiological adaptation to the sport. Aspects such as the athlete's physique, frequency and intensity of training, and the training program itself must also be taken into account since these external factors can lead to isolated changes [35].

These alterations would be very symptomatic of a cardiac problem under normal conditions. Finding these alterations using an EKG is not at all surprising though since many high-performance athletes enhance their exercise capacity and adapt their myocardium to these demanding circumstances [35]. The Cord (miocard) has physiologically adjusted to the hard exertion by making all the adjustments, including the ST elevation. The best course of action in these situations is to monitor the athletes, thus an EKG should be done on them every six months.

**Table 5.** EKG analysis results.

| Athlete | Initial Test Modifications | Clinical Significance | Final Test Modifications | Clinical Significance |
|---|---|---|---|---|
| A1 | Bradycardic rhythm | Increased effort tolerance | PR > 0.2 s | Atrioventricular block type 1 |
| A2 | Bradycardic rhythm | Increased effort tolerance | Bradycardic rhythm | Increased effort tolerance |
| A3 | Increasing the amplitude of the R wave in aVL > 11mm | Left ventricular hypertrophy | ST elevation from V2–V5 | Anterior stroke |
| A4 | PR > 0.2 s | Atrioventricular block type 1 | PR > 0.2 s | Atrioventricular block type 1 |
| A5 | Increasing the amplitude of the R wave in DI > 13 mm | Left ventricular hypertrophy | Increasing the amplitude of the wave R in DI > 13 mm | Left ventricular hypertrophy |
| A6 | Bradycardic rhythm | Increased effort tolerance | Increasing the amplitude of the R wave in aVL > 11 mm | Left ventricular hypertrophy |
| A7 | Bradycardic rhythm | Increased exercise tolerance | Bradycardic rhythm | Increased exercise tolerance |

Legend: PR segment—reflects the time delay between atrial and ventricular activations, ST segment—interval between depolarization and repolarization of the ventricles, R wave—depolarization of the main mass of the ventricle, aVL derivation—unipolar derivation of the members.

### 3.3. Astrup, Biochemical Blood Test and Urine Test Results

Table 6 lists all the parameters of relevance, together with their reference intervals and measurement units, that were altered in at least two of the seven patients. After evaluating the major results, the section on additional modifications discusses the other adjusted parameters that coincided with a certain issue.

**Table 6.** Biochemical interval values and measurement units [30].

| pO$_2$ | pCO$_2$ | SO$_2$ | Ca$^{++}$ | Cl$^-$ | LDH | CPK | P |
|---|---|---|---|---|---|---|---|
| 75–100 mmHg | 35–45 mmHg | 95–100 % | 4.61–5.33 mg/dL | 98–106 mmol/dL | 230–460 UI/L | 50–250 UI/L | +/- |

Legend: pO$_2$—partial oxygen pressure, pCO$_2$—partial pressure of carbon dioxide, SO$_2$—oxygen saturation, Ca$^{++}$—concentration of calcium ions, Cl$^-$—concentration of chlorine ions, LDH—lactate dehydrogenase, CPK—creatine phosphokinase, P—proteinuria.

Comparing the results obtained from the initial and final testing, the most significant changes are found in Table 7. This table contains both the values of the parameters and their dynamics represented schematically by arrows.

**Table 7.** Astrup, biochemical blood test and urine test results.

| Athlete | pO$_2$1 | pO$_2$2 | pCO$_2$1 | pCO$_2$2 | SO$_2$1 | SO$_2$2 | Ca$^{++}$1 | Ca$^{++}$2 | Cl$^-$1 | Cl$^-$2 | LDH1 | LDH2 | CPK$_1$ | CPK$_2$ | P1 | P2 |
|---|---|---|---|---|---|---|---|---|---|---|---|---|---|---|---|---|
| A1 | 66.6↓ | 72↓ | 37.5↔ | 35.9↔ | 93.2↓ | 94.3↓ | 4.2↓ | 4.28↓ | 111↑ | 108↑ | 518↑ | 318↔ | 215↔ | 166↔ | + | - |
| A2 | 62.1↓ | 69↓ | 34.5↓ | 37.5↔ | 92.6↓ | 93.7↓ | 4.2↓ | 4.69↔ | 110↑ | 105↔ | 448↑ | 301↔ | 439↑ | 236↔ | - | - |
| A3 | 54.7↓ | 60↓ | 37↔ | 39,6↔ | 87↓ | 90.2↓ | 3.7↓ | 4.1↓ | - | - | 325↔ | 289↔ | 182↔ | 143↔ | - | - |
| A4 | 66.1↓ | 66.4↓ | 47.1↑ | 34.3↓ | 92.2↓ | 93.2↓ | 4.1↓ | 4.1↓ | - | - | 503↑ | 470↑ | 210↔ | 205↔ | - | - |
| A5 | 71.7↓ | 82↔ | 34.8↓ | 42↔ | 93.8↓ | 94,2↓ | 4.27↓ | 4↓ | - | - | 637↑ | 348↔ | 649↑ | 329↑ | + | - |
| A6 | 60↓ | 74↓ | 48.7↑ | 46↑ | 89.7↓ | 91.8↓ | 4.47↓ | 4.23↓ | 110↑ | 108↑ | 367↔ | 325↔ | 254↑ | 251↑ | - | - |
| A7 | 54.5↓ | 68↓ | 44.1↔ | 41.6↔ | 86.9↓ | 93.1↓ | 4.6↓ | 4.25↓ | - | - | 362↔ | 294↔ | 468↑ | 385↑ | + | - |

Legend: pO$_2$1—initial partial oxygen pressure, pO$_2$2—final oxygen partial pressure, pCO$_2$1—initial partial carbon dioxide pressure, pCO$_2$2—final carbon dioxide partial pressure, SO$_2$1—initial oxygen saturation, SO$_2$2—final oxygen saturation, Ca$^{++}$1—initial calcium ion concentration, Ca$^{++}$2—final calcium ion concentration, Cl$^-$1—initial chloride ions concentration, Cl$^-$2—final chloride ions concentration, LDH1—initial dehydrogenase lactate, LDH2—final lactate dehydrogenase, CPK$_1$—initial creatinfosfokinase, CPK$_2$—final creatinesfokinase, P1—initial proteinuria, P2—final proteinuria, "+"—present, "-"—absent, ↓—decreased, ↔—unchanged, ↑—increased.

Following the mountain running competition, in the initial testing, all seven athletes recorded a low pO$_2$, reflecting hypoxemia, and in the final test, with the exception of subject A5, all subjects also recorded low pO$_2$ values, which denoted the increased need for oxygen to maintain aerobic effort [46]. In all subjects where both low pO$_2$ and pCO$_2$

were recorded, optimal respiratory failure was reported. In both tests, $SO_2$ was low for all subjects, signaling the tendency to overstrain the respiratory system during intense and lasting exercise. The decrease in the concentration of calcium ions was found in all seven subjects during the initial testing and six subjects in the final test, with the exception being subject A2. This denoted the intense metabolic consumption of calcium ions for the initiation and maintenance of muscle contraction, both for the striated muscle and the myocardial muscle [47]. Moreover, increases in chlorine ionic concentration were identified in three out of seven subjects in both the initial and final tests. These increases were slightly above the upper limit of the physiological interval, which most likely denoted slight dehydration due to effort. LDH is an intracellular enzyme that is found mainly in the kidneys, myocardium, skeletal muscle, liver and lungs. As a result of the effort performed, the increased LDH values signified the micro-rheumatisms of the skeletal muscle fibers that occurred after repeated contact with the running surface and excessive muscle contraction. In the initial testing, four out of seven subjects had increased post-effort LDH levels, namely, A1, A2, A4 and A5, while in the final test, only one subject recorded an LDH level above the physiological limit, namely, A4. The increase in CPK denotes the inflammation of the striated muscles, which is specific to the myopathy of effort after sustained and lasting effort. Four out of seven subjects recorded increased CPK after the initial testing and only three out of seven in the final test. The decrease in the LDH and CPK values after the final testing compared with the initial testing [48] may have meant a higher level of training from physical and tactical points of view, a better adaptation of the body to intense and lasting effort from the physical point of view in the second peak of the form compared with the first, and the improvement of the properties of the muscle fibers in order to prevent trauma, namely, elasticity and extensibility.

Proteinuria was recorded in subjects A1, A5 and A7 in the initial testing, with the values being recorded only as "present proteinuria" rather than expressed as the amount of proteins eliminated in the urine. The presence of a single "+" represents minimal proteinuria, "++" represents moderate proteinuria and the presence of "+++" represents marked proteinuria. Minimal proteinuria is physiological after intense effort and does not require further investigation [49]. However, performing a urinalysis in subsequent tests with the quantitative interpretation of proteinuria is preferable to provide a better interpretation.

Among other singular changes that occurred in the biochemical tests, an increased level of glutamic pyruvic transaminase (GPT) of 45 IU/L was detected in subject A1, which may have signified an overload of the liver due most likely to the metabolism of degradation products obtained after effort. Another change encountered in athlete A1 was minimal hypomagnesemia, which most likely signified the losses due to sweating. Subjects A2, A4 and A5 had low triglyceride levels (TGLs), both after the initial test and after the final test, which most likely signified an insufficient nutritional intake of fat, thus a revision of the diet was recommended.

Subject A3 recorded slight hyponatremia and hyperkalemia, which most likely signified slight dehydration after effort.

### 3.4. Factor Analysis Results

Furthermore, we decided to analyze each factor that influenced athlete performance measures as a dependent compound reflective variable: Performance2, made of time spent during the final race and results obtained during the NeuroTracker final test. We designed a model that had an independent construct as a formative variable: Physiologic2, made of the values of Min2—Bruce protocol duration (minutes), VO2/kg (mL/min/kg) for the final race, LATF2—lactic acid final test 2 (mL/min/kg) and LARF2—lactic acid rest test 2 (mL/min/kg).

The correlation coefficient value ($-0.607$) between the Performance2 and Physiologic2 variables (Figure 2) showed a medium negative correlation. We could affirm that when Physiologic2 had a smaller value, Test2 had a higher value. The path analysis demonstrated

that Physiologic2 had a rather strong impact on Performance2, which was the performance obtained by athletes in the final test.

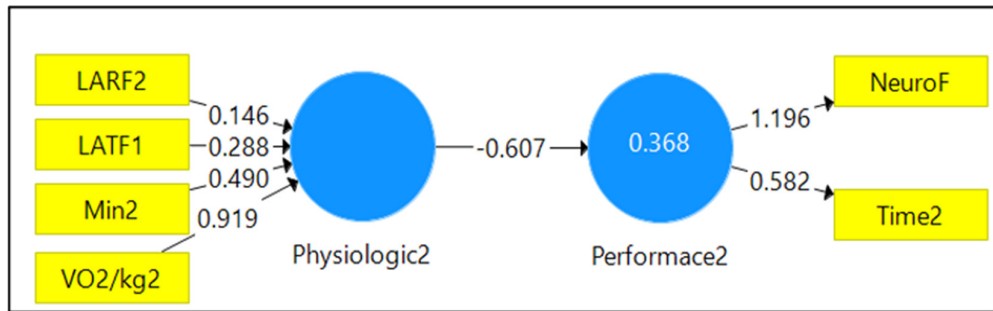

**Figure 2.** Path coefficients and loading factors.

We created a structural equation model using SmartPls 3.0 based on the aforementioned correlations. Since PLS-SEM is a non-parametric technique, the data did not need to adhere to specific distributional presumptions. To examine the significance of various outcomes, such as path coefficients, Cronbach's alpha, HTMT, and R-squared values, PLS-SEM instead uses a bootstrap approach [50,51].

Construct Reliability and Validity

SmartPls software provides many tests that can be used to ensure a coherent analysis and interpretation of data and to assume the research outputs. For example, the consistency of our model was grounded on the validation steps provided in Table 3 [52,53]. All the considered variables presented very high values for composite reliability: Cronbach's alpha and rho_A (>0.7—the bottom value authorized), as well as average variance extracted (AVE) (>0.5—the bottom value approved), meaning that the convergent validity could be assumed. These results empowered us to believe that our model was well determined and that our hypothesis (athletes' performance depends on physiological test values) was validated because all variables took values higher than the minimum threshold (Table 8).

**Table 8.** Model validation steps.

| Variable | CA | rho_A | CR | AVE |
|---|---|---|---|---|
| Physiologic2 | >0.7 | >0.7 1 | >0.7 | >0.5 |
| Sensory activity | 0.821 | 1.142 | 0.932 | 0.884 |

Legend: collinearity statistics were CA—Cronbach's alpha, rho_A —Composite reliability indicator, CR—composite reliability, AVE—average variance extracted.

Each construct's variance inflation factor (VIF) was used to evaluate the relevance of the variables. We excluded HRmax2 from our model due to high multicollinearity with $VO_2max$. There was no evidence of variable collinearity because the VIF was lower than 5—the acknowledged threshold (Table 9).

**Table 9.** VIF values.

| Variable | LARF2 | LATF2 | Min2 | NeuroF | Time2 | $VO_2max$ |
|---|---|---|---|---|---|---|
| VIF | 4.488 | 2128 | 1.271 | 1.941 | 1.941 | 2.793 |

*3.5. Statistical Analysis*

Performing a paired-samples *t*-test for the most significant variables gave the following results shown in Table 10.

**Table 10.** Statistical variables values.

| Pair of Variables | 95% CID | t-Value | *p*-Value | Result |
|---|---|---|---|---|
| Rank1-Rank2 | (−1.54,5.54) | 1.38 | 0.005 | RNH |
| $VO_2max1$-$VO_2max2$ | (−1.12,0.89) | −4.47 | 0.004 | RNH |
| NTi-NTf | (−0.5,0.2) | −6.59 | 0.004 | RNH |
| $pO_21$-$pO_22$ | (−12.5,−3.39) | −4.2 | 0.005 | RNH |
| $SO_21$-$SO_22$ | (−2.1,2) | −2.84 | 0.029 | RNH |
| LDH1-LDH2 | (116.4,99.17) | 3.1 | 0.021 | RNH |

Legend: CID—confidence interval of the differences, RNH—reject the null hypothesis.

From Table 10, it is observed that the null hypothesis was rejected for all the paired selected variables, which is a fact that confirms that the final test results were significantly better than the initial test. The subjects performed better in the final test compared with the initial one ($p < 0.05$ for the CID 95%) after performing all the medical investigations and modifying the training program daily according to the found values and they obtained better positions in the general rank of the specific mountain running race.

## 4. Discussion

The aim of the present study was to determine whether using preventive screening health tests on athletes, such as those discussed in this research, could positively influence sports performance in order to obtain better results at the races.

The main purpose of this category of selected and applied tests was to investigate a large area of possible modifications of the athletes' health statuses [54]. The second important purpose was recording and monitoring the obtained values of the main variables during the initial and final tests, and highlighting that their improvement led automatically to increased sports performance because the athletes obtained better results in the general ranking from the final mountain race compared with the initial one.

After performing systematized medical analyses and adapting the training program in order with them and in collaboration with their coach, most of the athletes registered a better basic lactate level before competing in the final test, a $pCO_2$ between the physiological limits, and higher $SO_2$ and $pO_2$ levels at the end of their efforts. Calcium ionic and chlorine ionic concentrations were closer to the physiological values after the final test and the detected LDH and CPK levels after effort were lower compared with the initial test for all the subjects. All these changes led without doubt to an improved physical shape, and thus, the subjects could more easily reach their top peak and compete in their best shape [55].

All subjects completed the Bruce protocol's final stage in the final test and all exceeded the initial Bruce protocol's maximum duration on the treadmill; furthermore, all of them recorded higher levels of aerobic capacity by reaching a bigger $VO_2max$ value. A significant correlation between the highest values of $VO_2max$ and the position in the general rank in the mountain running races was also observed, i.e., the higher the $VO_2max$ value, the better the results in the race were expected to be compared with those that registered a poor $VO_2max$. However, this correlation has its limits because to reach a high place in a race ranking, other requirements are needed, such as very good physical [56], tactical and mental training [57], as well as a good biochemical status of the organism. Thus, we can state that a superior $VO_2max$ value does influence obtaining maximum results during a race in a good way [58], but it does not automatically determine a higher place in the race rankings.

Regarding electrolyte consumption, it was observed that the main changes were calcium ionic consumption and sodium ionic consumption, and higher levels of chlorine were noticed at the end of the effort. These led us to the fact that better hydration during long mountain races is required [59], as well as using a better vitaminization protocol before competing [60]. Subjects should revise their nutrition under strict recommendations from a

nutrition specialist in order to reduce the damage produced by effort and to keep the main biochemical modifications within the physiological range.

The path analysis showed that Performance2, which was the performance attained by athletes during the final exam, was significantly influenced by Physiologic2. Therefore, we could state that the physiologic test results influenced the athletes' performance [61]. This verified other studies that showed that NeuroTracker training improved sports performance in other situations, such as high-intensity interval training [32] or sports such as archery [62], basketball [63] and women's soccer [64].

Promoting and utilizing medical investigation methods to evaluate an athlete's health status should be a worldwide concern with the aim of protecting them from the multiple health issues that they might develop [65] due to the extensive stress that sports performance exerts over their body [66]. Although in some less developed countries, the usefulness of these tests may be partially misunderstood or may generate costs that are difficult to cover [67], their use is very important because it can prevent typical athlete ailments, such as anemia, Brugada syndrome, sudden death, burnout syndrome, fractures and muscle ruptures [68].

## 5. Conclusions

The findings of this study highlighted that using medical modern technology, such as the Radiometer automatic blood gas analyzer, Medonic M-Series Hematology Analyzer and Lactate Pro-2, was easy and efficient, offering real values of the most significant biochemical parameters in order to give a short overview over each athlete's health status and to modify the training program of the day according to those values. Despite the fact that this technology might be expensive and requires a medically trained person to perform the tests and to interpret them, they are without any doubt the most significant contribution that can be made to improve sports results and an athlete's health status and prevent the most usual and known sports ailments.

Performing the paired-samples *t*-test demonstrated that the final test results were better than the initial ones, which showed that the subject improved their results in the final test, and thus, the reached sports performance was higher. This was also confirmed by the registered values in the final test for the main parameters: $VO_2max$, LDH, CPK, $SO_2$, P, NeuroTracker and, most importantly, Time2.

The NeuroTracker training demonstrated that perceptual–cognitive skills were perfectly trainable and it could improve sports performance by developing attention and concentration capacities.

**Author Contributions:** Conceptualization, V.M., I.G. and R.-S.E.; methodology, V.M., I.G. and R.-S.E.; software, V.M., I.G. and R.-S.E.; validation, V.M., I.G. and R.-S.E.; formal analysis, V.M., I.G. and R.-S.E.; resources, V.M., I.G. and R.-S.E.; data curation, V.M., I.G. and R.-S.E.; writing—original draft preparation, V.M., I.G. and R.-S.E.; writing—review and editing. V.M., I.G. and R.-S.E.; visualization, V.M., I.G. and R.-S.E.; supervision, V.M., I.G. and R.-S.E.; project administration, V.M., I.G. and R.-S.E. All authors read and agreed to the published version of the manuscript.

**Funding:** This research received no external funding.

**Institutional Review Board Statement:** Ethical review and approval were waived for this study due to the fact the respondents gave their consent to use the research results.

**Informed Consent Statement:** Ethics approval and consent to participate was granted. After receiving information on the objectives and procedures of the study, the participants signed an informed consent form, which complied with the ethical standards of the World Medical Association's Declaration of Helsinki (2013). The study was approved by the ethics committee of San Jorge University with registration number: 009-18/19.

**Data Availability Statement:** The datasets generated and analyzed for this study can be requested via correspondence with the author at iulia.gainariu@student.unitbv.ro.

**Acknowledgments:** Technical support offered by the National Sports Research Institute was very important in accomplishing this research.

**Conflicts of Interest:** The authors declare no conflict of interest.

## List of Abbreviations

$VO_2$max—maximum aerobic capacity; pH—potential of hydrogen; $CO_2$—carbon dioxide; $pCO_2$—partial pressure of carbon dioxide; $pO_2$—partial pressure of oxygen; EKC—electrocardiography; MRR—mountain running race; EDTA K2—dipotassium ethylenediaminetetraacetic acid; EDTA K3—tripotassium ethylenediaminetetraacetic acid; mL/min/kg—milliliters/minutes/kilogram; GPT—glutamic pyruvic transaminase; Mg—magnesium; Ca—calcium; Na—sodium; K—potassium; Time1—time spent during initial race (milliseconds); Time2—time spent during final race (milliseconds); Min1—Bruce protocol duration (minutes) vs. Min2—Bruce protocol duration (minutes); $VO_2$max (mmol/L for initial and final race and HRmax for initial and final race; $pO_2$1—initial partial oxygen pressure; $pO_2$2—final oxygen partial pressure; $pCO_2$1—initial partial carbon dioxide pressure; $pCO_2$2—final carbon dioxide partial pressure; $SO_2$1—initial oxygen saturation; $SO_2$2—final oxygen saturation, $Ca^{++}$1—initial calcium ion concentration; $Ca^{++}$2—final calcium ion concentration; $Cl^-$1—initial chloride ions concentration; $Cl^-$2—final chloride ions concentration; LDH1—initial dehydrogenase lactate; LDH2—final lactate dehydrogenase; CPK1—initial creatinfosfokinase; CPK2—final creatinesfokinase; P1—initial proteinuria; P2—final proteinuria; CA—Cronbach's alpha; CR—composite reliability; AVE—average variance extracted; LATI 1—initial lactate Test 1 vs. LARI 1—initial rest 1 lactate test; LATF2—final lactate test 2 vs. LARF2 final lactate rest test 2; NeuroI and NeuroF—NeuroTracker test scores for the initial and final races.

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
