# Peer review of "Implementing Modern Technology for Vital Sign Monitoring to Enhance Athletic Training and Sports Performance"

_sustainability, doi:10.3390/su15032520_

Round 1

Reviewer 1 Report

Nice paper. It should be published

Author Response

Dear reviewer,

Thank you so much for your opinion.

Reviewer 2 Report

Introduction.

The introduction presented correctly supports the research. In this section, the authors have used current literature relevant to the subject developed in the document.

Materials and methods.

Participants.

This section presents one of the main problems of the research, the sample size is very small.

It is necessary for the authors to explain the type of sample used and what were the criteria for inclusion and exclusion of the sample members.

The sex of the participants is not indicated either, as this data is important for the evaluation of the results obtained in the different tests presented in the research.

Procedure.

It is not clear how much time elapsed between competitions and whether this could have an influence on the results obtained in the tests. How much time elapsed between the last competition and the second data collection?

Why was an intermediate data collection not scheduled?

Materials and testing.

The definition of the different tests performed is correct. Likewise, the objective pursued by the authors with their application is clear.

Statistical Analysis

What normality test has been applied?

Results

The results are in line with the research objective and are clearly presented. The tables and graph help the understanding and interpretation of the results.

However, some more results should appear, such as the results concerning the normality test.

It would also be interesting to differentiate by gender.

Discussion

It is suggested to introduce more references to similar studies and the results obtained in order to be able to make a comparative analysis with your research.

Introduce more bibliographical references in this regard.

Conclusion

In accordance with the text initially presented.

Bibliography

It presents a very high percentage of current references.

Author Response

Dear reviewer,

Thank you so much for your kindly recommendations.

Please see the attachment with my answer.

Thank you!

Reviewer 3 Report

Dear authors:

Introduction:

-Please include more recent references in the introduction. 

-Kindly include the following reference in the introduction section, after including a short paragraph about the nutrition knowledge of mountain running athletes:

Serhan, M., Yakan, M., & Serhan, C. (2022). Sports nutrition knowledge translates to enhanced athletic performance: A cross sectional study among Lebanese university athletes. Nutrition & Food Science, https://doi.org/10.1108/NFS-07-2022-0228 (vol. ahead-of-print no. ahead-of-print).

Materials and methods: 

-Why participants are only men? Especially that recruitment of participants was done by invitation. 

-Are the procedures applied and described in Table 2 based on a reference?

-in the section of materials and testing: the following sections should be shortened to the maximum: mountain running race, effort test, lactate analysis, neutrotracker, EKG test, MicroAstrup, biochemical blood and urine test...

Discussion:

Kindly enrich the discussion with references. Please make the discussion of findings more coherent and balanced. 

Author Response

Dear reviewer,

Thank you for your kindly recommendations.

Thank you!
